# Proposals of Processes and Organizational Preventive Measures against Malfunctioning of Drones and User Negligence



Mircea Constantin Șcheau [1,2], Monica Violeta Achim [3], Larisa Găbudeanu [4,*], Viorela Ligia Văidean [3], Alexandru Lucian Vîlcea [5] and Liliana Apetri [6]

1   Institute for European Studies, Babeș-Bolyai University, 400084 Cluj-Napoca, Romania
2   Faculty of Automation, Computer Science and Electronics, University of Craiova, 200585 Craiova, Romania
3   Department of Finance, Faculty of Economics and Business Administration, Babeș-Bolyai University, 400084 Cluj-Napoca, Romania
4   Faculty of Law, Babeș-Bolyai University, 400084 Cluj-Napoca, Romania
5   Cybernetics Faculty, Bucharest University of Economic Studies, 010552 Bucharest, Romania
6   European Cybersecurity Organisation, 1000 Brussels, Belgium
*   Correspondence: larisa.gabudeanu@ubbcluj.ro

**Abstract:** Drones have been included in more and more activities in various domains, such as military, commercial and personal use. The existing legislative framework insufficiently addresses the responsibility and preventive measures angles in case of vulnerability exploitation and negligence in drone usage. Such aspects can be addressed by the industry in technological processes and standardization. These are especially important aspects given the high impact that misuse of drones can have on individuals, property and buildings within the flight zone when the drone is misused. The aim of this research paper is to investigate how these elements are viewed in existing legislation and by individuals, while taking into account the technical specifics and the stakeholder ecosystem of drone usage. In this respect, we use a complex questionnaire which was sent to a final number of 233 respondents pertaining to firms specialized in IT, legal and cybersecurity. The responses have been analyzed from a qualitative and quantitative perspective. Our results highlight the areas of improvement in the existing standardization and find the followings: (1) stakeholders across the drone ecosystem are viewed as having a shared liability in certain use cases, (2) preventive measure implementation should be dispersed across the stakeholders of drone usage and (3) automation of prevention measures is considered more useful in case of malfunctioning or misuse of drones rather than user manual intervention. In addition, we make proposals to accommodate new policy requirements for the above use cases. The results of this research paper assist policy makers in improving existing standardization framework and technological processes concerning drone usage, but also stakeholders of the drone ecosystem in generating increased trust of the drone users. Further, this research paper can also assist drone software and hardware producers in calibrating their products to ensure trust of the users. In addition, trust in the use of drones for commercial and personal purposes is increased through standardization and proper approaches for situations that may cause damages to drones and to third parties.

**Keywords:** vulnerability management; user centric security; security by design; digital responsibility

## 1. Introduction

In recent years there has been a development in the use of drones for military, commercial and personal use. Generally, a higher degree of digitization can result in a higher degree of risks [1]. Such risks include use cases that can fall under three categories: user actions leading to economic damages, successful cyber-attacks and errors/malfunctioning of the drone (either software or hardware part). The below sections explore each of these three categories with details on potential use cases for each of them, with technical details on the cyber threats. Thus, there are two types of potential generators for economic damages:

internal and external ones [2]. The internal ones refer mainly to errors, misconfigurations or bugs in the software or hardware of the drone. The external ones refer mainly to actions by persons/entities on the drone in order to generate the economic damages. These external factors can include the user itself or a third party acting willfully to generate economic damages.

These uses of new technologies have tested the existing legislation in terms of liability and preventive measures and have paved the way to new interpretations of existing legislation as well as the need for additional legislation tailored to the new technological landscape. The general rules on tort law together with specific rules of product liability have proved insufficient to address appropriately all use cases in the drone ecosystem [3,4].

In this context, there have been debates on certain liability aspects in case of negligence in using drones and on malfunctioning of drones stemming from existing vulnerabilities within them [5]. Such research papers focus on the legislative side of these aspects. From a practical perspective, at industry level, through standardization, organizational processes are also essential in order to create a common approach for roles and responsibilities of the stakeholders involved in the drone lifecycle.

*This research paper focuses on these specific issues of liability in order to identify the manner in which this should be addressed given existing legislation and industry standards, how this is viewed by individuals using drones and proposals for industry standards reflecting the roles and responsibilities of each stakeholder. In addition, the research paper goes further and proposes preventive technical measures to be implemented by the relevant stakeholders identified as responsible in a particular situation.*

In view of addressing the above, a qualitative analysis of existing regulations is performed, taking into account the technical particularities of drone usage and the stakeholder ecosystem related to drones.

A quantitative approach is used to validate the finding through a questionnaire aimed at industry professionals in the fields of IT, legal and cybersecurity addressing liability and preventive measures aspects in order to obtain the view of the respondents on the approach they consider appropriate on these topics.

Three main conclusions can be drawn from the results of this research paper:

The civil liability in case of negligent actions or inactions is viewed as a shared responsibility between multiple stakeholders across the drone ecosystem. This is due to the role that each of them has in creating and setting up the drone for usage.

This ties in with the second conclusion, which refers to preventive measures. These are also seen as a shared responsibility across the drone ecosystem, with each stakeholder being considered essential in implementing certain types of prevention measures.

The third conclusion is linked to this as well. Respondents are expecting an automated response from the drone software in case of malfunctioning and even to prevent negligent maneuvers performed by the drone user.

*These three conclusions can lead to improvements in recent draft standards in the field of internet of things solutions, including NIST SP 800-82 r3 and the proposed EU Cyber Resilience Act, as further detailed in this paper.*

*The research article brings together in an innovative manner the technical use cases that can generate damages through negligence and translates these into standardization steps within the industry that can be taken to prevent and to remediate this situation, while taking into account the role that each stakeholder in the drone ecosystem can play in this respect.*

*Thus, the innovative approach of analyzing existing specific or general legislative framework, proposing approaches for the above negligence use cases and validating such approaches through quantitative means is proposed.*

The research article analyses the above aspects in terms of technological process, need for standardization of the technical and organizational concerns related to the scope of this article from a prevention perspective.

Further, the research addresses only civil drones and not military or institutional drones and takes into account for analysis the current civil legislation at the level of the European Union and in civil law principles in such countries.

The rest of the article is organized as follows. Section 2 presents the literature review in the field of drones from two angles, standardization and cyber security. Further, Section 3 concentrates on hypotheses and objectives. Section 4 describes data and methodology. Sections 5 and 6 go into additional insights with respect to the results and proposals for adjusting standardization based on these results. The final section presents the conclusions of our research, the limits and future research directions.

## 2. Literature Review

We have targeted the literature review and we introduced in addition a correlation map between keywords from the domain theme as detailed in Figure 1, in order to highlight the importance of the research works that way.

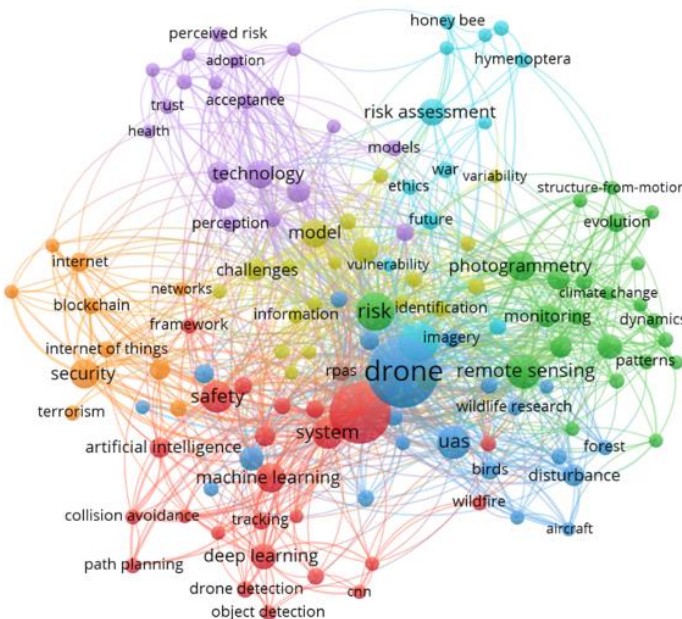

**Figure 1.** Map of articles relevant for the keywords using the Web of Science database.

The articles in this map reflect the dual directions of research on this topic, the organizational and standardization one (e.g., framework, risk assessment, perception, acceptance, monitoring) and the technical implementation thereof (e.g., vulnerability, path planning, collision avoidance).

Most of the existing studies on drones refer to their mode of operation [6] or their usefulness in various fields [7]. Of course, these studies have led to a better understanding of how to handle these kinds of devices, but also make an overview of the diversity of opportunities for using them. The literature related to damages and civil liability in case of drone usage is rather limited.

From a standardization perspective, the literature on drone legal issues usually analyses the drone authorization process and the standardization thereof across a specific area (e.g., the European Union), including multi-disciplinary research papers such as Konert and Balcerzak [8]. This has been a topic for the last decade, given the fact that the use of drones was still an emerging technology usage.

However, there are certain research papers in the common law space, with emphasis on US literature [9], whereas for the civil law countries, there is less literature on this topic [10]. Other studies refer to drone regulation as a good example for emerging technology governance. Studies of cases of drone operators' prosecution can also be found, such

as the article powered by Skytango, a company that addresses Enterprise-type customers who manage teams of drone pilots in various fields of activity [11].

In Table 1 below, we have done a brief presentation of the top five cases of this topic in order to give a little insight into what it means to be non-compliant with local drone regulations of a country. From last years' events, these are the cases that have received the highest fines/prison sentences. Most common causes consist of the use of drones over areas where UAVs are prohibited or unregistered.

**Table 1.** Top 5 cases of drone operators' prosecution.

| Country | Fee Amount | Cause | Conclusion |
|---|---|---|---|
| United States of America | USD 200,000 | The event was judged in October 2015. The proposed fine in the first round was USD 1.9 million, offsetting 43 illegal drone flights over New York and Chicago between 2012 and 2014. SkyPan has been accused of operating 65 drones without communication tools without being registered or licensed. Following a SkyPan agreement with the FAA (January 2017), the fine was reduced to USD 200,000, plus USD 150,000 if they continued to violate the rules. | The FAA mentioned that SkyPan used the aircraft outside of the recommended approaches, which can lead to danger for individuals and assets. |
| United Kingdom | GBP 1800 | Nigel Wilson, a drone enthusiast, illegally flew nine times over football stadiums in England and over buildings in central London (Big Ben, Buckingham Palace). The distance from them was about 50 m. He revealed his actions by posting them on YouTube. He was fined GBP 1800 for his offenses under the Air Navigation Order (2009). | Wilson endangered the public through the use of drones over building and busy areas. |
| United Kingdom | GBP 1125 | Richard Brunner illegally operated a flight with a drone over the set of a promotional spot above Hyde Park. The distance was 10 m and he was fined GBP 1125. | The area is restricted to fly over and prior permission was required to fly over this area. |
| Netherlands | EUR 8000 | The famous violinist and conductor André Rieu operated a flute with a drone over a stage where a show was taking place at the Vrijthof in Maastricht. The area was central, so piloting drones was illegal. In total, 12,000 people were in the show space. The amount of the fine was not made public, but according to the press it would have amounted to EUR 8000, being the largest fine granted by the Dutch authorities. | Improper usage of drones—Flying a drone above an inner city in the Netherlands is prohibited. |
| China | 18 months in jail | A Beijing UAV Sci-Tech Co staff member was sentenced with 18 months in jail, in 2015, for using a drone of the company that disrupted commercial flights. | The staff member from Beijing UAV Sci-Tech Co endangered public safety. |

Source: Author's processing, based on a Skytango article.

In certain research papers, the liability according to general tort law principles is explored, without emphasis on analysis of specific use cases in this respect. General tort law, either in common law or civil law countries entails the existence of an action (or inaction), the existence of damages incurred by an individual (or a legal entity) and a casualty liaison between the two elements. In civil law, a person may be liable for his/her own actions or for the actions of an asset it owns or otherwise controls. In the case of the complex drone ecosystem encompassing stakeholders such as drone hardware producer, drone software producer, drone integrator, distributor, authorities, certification bodies, maintenance providers, in certain cases, it is difficult to pinpoint the entity whose actions fulfil the tort law conditions in terms of action, causality and consequence [12,13].

Other research papers make reference to specific legislation, such as EU legislation concerning liability for products. Nevertheless, this view limits the involvement of other stakeholders in the drone ecosystem [14]. Furthermore, it focuses on the post factum situation rather than also addressing the preventive angle. Other studies reflect the use of UAVs under international law, but only referring to a certain country, namely The

Netherlands. Further research focuses only on legal consideration and not bridging the gap towards a multi-disciplinary analysis of the use cases [15]. In addition, certain studies also go in the direction of ethics and standardization aspects [16].

From a technical perspective, there are certain studies that explore different manners of preventing damages or losing control of the drone [17] because of negligent actions of the user and also detecting and addressing malfunctioning of the drone while in flight [18]. Also referring to a technical view of the issue, we can identify certain studies about the evaluation of the risk of unmanned aircraft ground impacts [19] and the risk-based approach that can be taken to approach this [20]. Other studies refer to mobile facilities (services/applications) that use air infrastructure, but also their related vulnerabilities, in order to optimize their configuration and UAVs efficiency, and in order to reduce the occurrence of unforeseen events. In addition, there are studies that provide a taxonomy of cyber-threats for drone and cyber-mitigations that can be implemented, without going into detail on the technological process for addressing such cyber-threats and cyber-mitigations within the circle of stakeholders involved in the drone creation, maintenance and usage [2,21]. However, these are not currently correlated with legal obligations and the rest of the existing legal framework or standardization within the industry.

In September 2020, the Australian Government Civil Aviation Safety Authority published the Micro and excluded remotely piloted aircraft operations guide in order to assist drone users to understand and apply drone regulations and safety rules before their first flight, before every flight and during every flight, also taking into consideration the no-fly zones [22]. This is a step in the direction of standardization of prevention organizational mechanisms. Nevertheless, from a process perspective, additional steps have to be taken for the standardization in this respect.

Based on the information presented by Dedrone, the Airspace Security Insight Report 2022, in 2021, the U.S Federal Aviation Administration (FAA) launched the Airport Unmanned Aircraft Systems Detection and Mitigation Research Program [23], a program to test counter-drone technology at five U.S. international airports. The objective of this program was to make airports safer for passengers and manned aircraft from drone threats. Additionally, in 2021, INTERPOL carried out three-day exercises to evaluate and test drone countermeasures in a secure airspace environment through the detection, tracking and identification of drones and their pilots. These systems are emerging as essential elements in ensuring the security of airports, airspaces and protecting no-fly zones above cities, prisons and critical infrastructure. Thus, this presents technical aspects to consider and implement. Nevertheless, in a civil context, the process standardization and organizational aspects are relevant to ensure proper and efficient implementation.

In June 2022, the Department of Defense Responsible AI Working Council published the US Department of Defense Responsible Artificial Intelligence Strategy and Implementation Pathway paper, which presents the industry's best practices for AI development and applying tools, and technologies to identify and mitigate risks as they relate to AI Capability throughout the AI product's lifecycle [24]. This also entails suggestions of technological processes that can be applied in the context of drones and prevention of certain malfunctions or actions of users.

Given this existing research in this field, our research paper focuses on bridging the gap between liability determination and setting-up correlative prevention measure mechanisms within the technological process and standardization environment. The current research paper is part of a complex research project concerning preventive cyber security measures for drone usage, which also includes [25,26].

## 3. Hypothesis and Objectives

The research in the above section analyzes the risks and liability concerning regarding drone usage from only one angle, legal, without analyzing the interplay between multiple angles. The production and use of drones generates certain cyber vulnerabilities for which there are solutions of remediation and prevention from both a technical or standardization

and technological process framework perspective. For this purpose, the existing literature does not cover the analysis of technical aspects and their impact from a standardization and technological process framework perspective.

This research paper was built on the above literature and explores further and in detail the interplay between technical and legal angles, leading to reducing the vulnerability exploitation and increasing the trust of clients from various sectors of the economy and for individual use.

In this respect, the scope of this article is to identify potential gaps in standardization/technological process framework and the manner in which these can be addressed (in case of negligence, for preventive measures and liability). In view of identifying relevant technical measures to address the identified gaps, this research paper analyzed the best practices that can be implemented for prevention.

The research paper reflects the respondent's opinion on points concerning the damages incurred because of misfunctioning and negligence, together with proposed preventive measures, and which is reflected through the following objectives:

Objective 1 (O1)—Identifying the opinion in terms of responsibility in case of incident caused by negligence of the drone user and malfunctioning of the drone.

Objective 2 (O2)—Identifying the mechanisms considered useful for prevention of damages in case of negligence from the drone user and malfunctioning of the drone.

**Hypothesis 1 (H1).** *(regarding O1)—Drone pilots are considered liable for incidents caused by negligence in flight.*

**Hypothesis 2 (H2).** *(regarding O2)—Periodic training of drone pilots is considered useful to prevent negligence incidents from occurring.*

**Hypothesis 3 (H3).** *(regarding O1)—For hardware deficiencies, the majority of respondents view the integration entity for the hardware and the software of the drone as responsible.*

**Hypothesis 4 (H4).** *(regarding O2)—Stakeholders early on in the drone lifecycle are considered responsible for setting in place proper preventive measures against malfunctioning and/or user actions.*

**Hypothesis 5 (H5).** *(regarding O1)—Automated mechanisms are preferred in terms of preventive measures rather than manual actions to be taken.*

**Hypothesis 6 (H6).** *(regarding O1)—Drone software producers are considered responsible for any vulnerabilities in the software, either identified as a vulnerability at the outset or after a period of time since the drone has started to be used.*

**Hypothesis 7 (H7).** *(regarding O2)—A combination of real-time preventive security solutions relating to the drone and periodical audits yield the most effective preventive measures.*

## 4. Data and Methodology

For the analysis performed in this research paper, we created and used a questionnaire using QuestionPro. This questionnaire assists with identification of responsibility and preventive measures in drone usage. The questionnaire was developed in both Romanian and English and was distributed to professionals over the internet, through e-mails and social media, including LinkedIn, Twitter and Facebook. The respondents include professionals in the relevant field between May and July 2021. It contains 17 questions, with 2 questions used to categorize the respondents' sample. This research paper concentrates on specific questions relevant for the topic analyzed herein.

A filtering process was performed to check invalid respondents and incomplete questionnaire responses. After this process was completed, a total of 233 questionnaires were fully completed, in English and in Romanian by persons within specialized IT, legal and

cybersecurity companies within Europe and Northern America. For the English version, the distribution is detailed in Figure 2. Because of the language of distribution, the interpretation of the responses can slightly differ. A significant number of respondents were from Romania (44.12%).

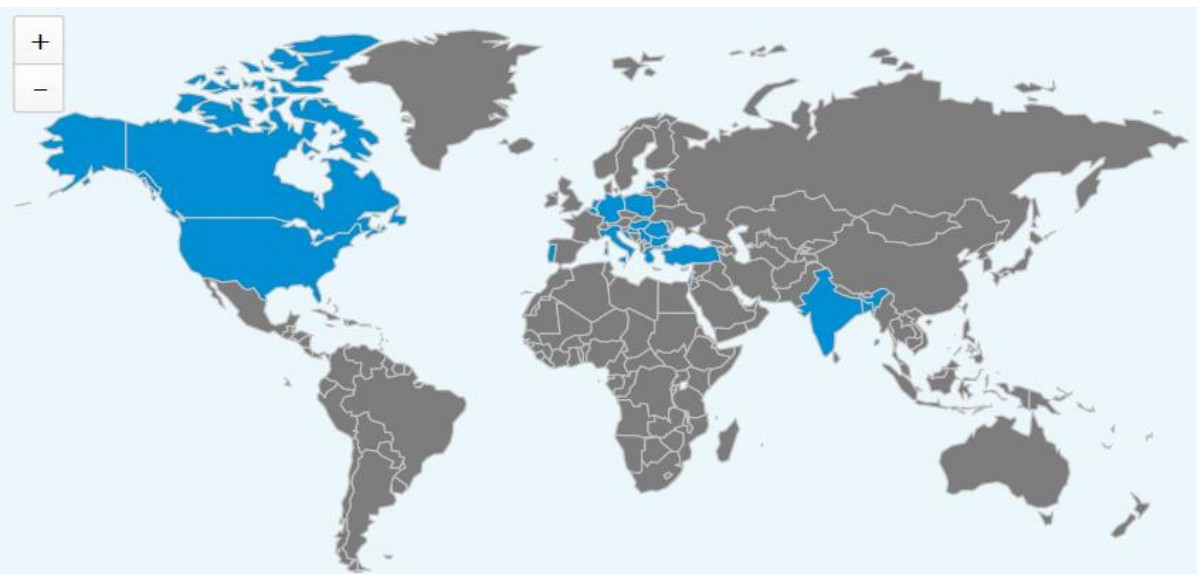

**Figure 2.** Distribution of respondents from a geographical perspective (blue represents the countries of the responders). Source: QuestionPro; analytics derived from the questionnaire and provided by the authors.

This research paper focused on two use cases: negligence actions of the drone users and malfunctioning of the drone. Both use cases are analyzed from two angles: responsibility in case of damages and prevention measures that can be implemented by one or more stakeholders involved in the drone's usage.

The responses to the questionnaire are analyzed from a distribution perspective, by analyzing the frequency of individual values and the frequency distribution.

Further, relevant responses are analyzed in terms of the central tendency of a distribution by calculating the median to analyze the distribution values.

In addition, the questions have been designed to address the same topic from a dual perspective, liability for damages and positioning of stakeholders to implement preventive measures for such damages. Thus, the analysis entails the correlation of the liability and preventive measures responses for each such pair of questions in order to identify the matching of responsibility of relevant stakeholders.

The analysis from a qualitative and quantitative perspective is complemented by the recommendations in terms of standardization and technological process framework changes. The focus has been to match the results of the quantitative research with the existing draft standards NIST SP 800-8 r3 Guide to Operational Technology (OT) 3 Security and draft EU Cyber Resilience Act (Proposal for a Regulation on horizontal cybersecurity requirements for products with digital elements).

The questions on responsibility attempt to identify the opinion of the respondents about responsibility in certain scenarios relevant for the research. From an economical viewpoint, this also reflects the view of damage covering.

The questions concerning preventive measures that can be implemented analyze the view of the respondents on certain policy proposals for prevention of cyber-incidents.

## 5. Results

This section details the results of the questionnaire analysis. The two angles detailed in this section are of significant economical relevance for both drone users and drone

producers alike and are highly dependent on standardization and proper description of technological processes. These are typically the use cases encountered throughout the lifetime of the drone. Thus, the two main aspects concerning damages caused by negligence, the actions taken (or not taken) by users that lead unintentionally to damages and misuse of drones and the malfunctioning caused by various vulnerabilities or misconfigurations of the drone software or hardware, are discussed.

### 5.1. User Actions Resulting in Economic Damages

The first type of threat is the one related to drone users. The users can act with intent (willful misconduct, generally named insider threats in cyber security) or with negligence.

One of the most significant drone incidents by industry causing economic losses and operational disruption was in December 2018, at Gatwick, the second business airport by total passengers traffic in the UK, resulting in a 36 h shutdown action, taken in order to avoid collision of two drones with aircrafts. This action cost the airport at least GBP 15 M (USD 19.8 M), with 140,000 passengers and 1000 flights affected. The incident was provoked by a couple of drone enthusiasts who lived near the airport who were arrested because they endangered the safety of operations or persons by using drone within 1 km of the airport boundary. After this event, Gatwick installed a permanent drone detection system to ensure the continued safety and resilience of the airport.

The two types of actions mentioned above resulted in different qualification of liability according to the general civil liability legal provisions and may also be mitigated differently from a standardization and organizational perspective. This use case ties in with H1 whereby drone pilots are considered liable for incidents caused by negligence in flight and with H2, whereby periodical training of the drone users can decrease such negligence damages cases.

One question in the questionnaire was "Who is responsible when damage occurs due to the negligent flying of drones?". We can find the answers in the legend in Figure 3.

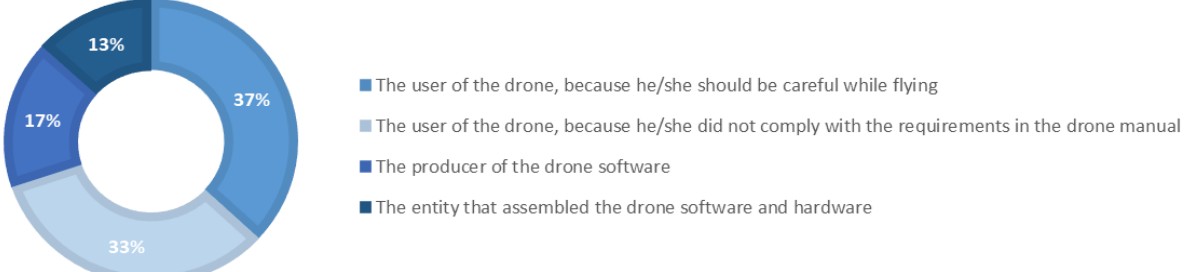

**Figure 3.** Question responses—responsible for damages. Source: Author's processing.

From the proportions found in Figure 3, we can see that the respondents consider the drone user as guilty in these situations. The first reason is negligence during the flight, and the second is non-compliance with the user instructions of the device. In this case, we can conclude that the drone provider is not responsible for such accidents, because it provided all the necessary elements for proper operation, but the user was careless and did not comply with them.

In the questionnaire, two hypotheses were analyzed. Firstly, the duty of care of the user (a general one in terms of flying and similar to driving a car). This was considered by around 37% of respondents as an essential obligation of the user, with the majority of such respondents located in Europe. Secondly, the obligation of a user to comply with requirements for using the drone (as specified by the drone producer). This was considered by 33% of respondents as essential, with the majority of such respondents located in Europe. Thus, the majority of the respondents view negligence as solely a user error. It is interesting that 30% of the respondents mentioned that the drone producer (or the entity the assembled it) is liable. This can indicate that respondents view the drone producer as an entity that can save the user from his/her own actions, by anticipating certain ill uses of the drone

(not in compliance with its purpose, its specifications or requirements for use) and include technical mechanisms to prevent negative consequences in certain cases (those that can be anticipated and remedies or stopped).

Therefore, *H1 is validated* with 70% by the views of the respondents, with almost equal reasoning for the liability for the drone user: carefulness when flying and proper knowledge of drone flying (including crisis management).

This is in line with the identified lack of proper preventive steps on the side of the users detailed in this section. Thus, as mentioned above, three aspects to implement in order to prevent accidents by negligence are: (i) the drone hardware and software producer to include details on the manner in which the drone can operate, (ii) a mechanism is set up for initial and periodical certification of the users with hands-on drone driving in specific situations and (iii) updating the user about new vulnerabilities and periodical verification of the drone's status. These aspects can add to the overall costs of drone production (and maintenance) and also additional costs and effort on the user side. In order for this approach to be successful, proper awareness on flight risks is also important, together with proper rules in place to accommodate the above from a technical and operational perspective.

Strictly from a legal tort law perspective, in such a case, it may be argued that the user is not at fault for the damages incurred by the affected individuals, given the lack of intent of the user [27]. This qualification greatly depends on the type of action or inaction performed by the user. In certain cases, even actions/inactions performed with negligence can be considered as sufficient for triggering civil tort law (e.g., noticing that the device on which the drone operating software runs may be infected with malware, but the drone continues to operate). Thus, in practice, this can be analyzed on a case by case basis. Nevertheless, more clarity on the actual liability can be established through clearer standardization requirements for users and the steps to be taken by them throughout the lifecycle of the drone, including maintenance and before each flight [28].

In addition, this case is closely related to the one in Section 5.1. Given the high impact that the drone incident can have on personal or public assets or, even on individuals, it is essential to include preventive technical and organizational security measures in terms of potential attacks, including drone self-check (including the self-check of the software used for operating the drone on the user device) before commencing a flight. This can be coupled with additional training for users.

In this context, from a technical perspective, additional automation to identify abnormal behavior or to double-check user instructions can be valid options. Such options have been analyzed in the literature from a technical perspective, without additional supply chain or lifecycle context [29,30]. Of course, there are multiple angles to approach this aspect and the angles should have in mind all layers of drone usage, from communication lines, sensor data to avoid collision, location, etc.

Organizations should use anti-drone solutions to protect people, property and information from drone-based threats. In order to protect people, in the case of controlling the drone via a tablet, computer or mobile device, it is absolutely necessary to maintain a malware-free environment. Two services can be useful in protecting the controlling device: a real time antivirus program and anti-theft protection for the device and VPN for station protection. By subscribing to VPN, the drone user connection becomes encrypted and internet usage is anonymous. Another method to protect the drone against hackers' control is with the aid of personal behavioral changes. A consistent path may be used by hackers to learn where the drone will be and target it. Keeping the drone in view can let users know if something is going wrong.

Drone monitoring technology is available for detection (a radar that can detect drone), classification (ability to separate drones from other types of objects), identification (identify the particular model of the drone), locating, tracking and alerting the users that a drone is present. There are four types of drone monitoring: Radio Frequency (RF) Analyzers –they are used to detect radio communication between a drone and its controller, acoustic sensors (microphones) that can detect the sound made by a drone and calculate a direction,

optical sensors and video cameras, which provide visuals on the drones and their potential payload, and can record images as forensic evidence, which can be useful in eventual prosecution. The drone detection radar sends out a signal and receives the reflection, measuring the direction and position of an object.

Drone countermeasure equipment can be:

- Ones that physically destroy the drone, but current regulations forbid the use of technologies to be used for neutralizing drones.
- radio frequency jammers that transmit a large amount of radio frequency energy toward the drone in order to mask the controlling signal.
- high power microwave devices that generate an electromagnetic pulse capable of disrupting electronic devices.
- high energy lasers thatdefeat the drone by destroying its structure and/or the electronics.

Thus, from a process perspective, the industry standardization and technological processes should focus more on the roles and responsibilities of the above mentioned stakeholders [31,32]. This can be reflected throughout the supply chain of drones.

Another question of the questionnaire was "In case of negligent flying, what are the relevant measures for preventing damages? (one to five scale, one meaning total disagreement and five meaning total agreement)".

Table 2 reveals the number of answers for each existing option, on a one to five scale. The last column shows the average score, similarly calculated like the one from the previous question. The greatest average score belongs to the drone automatic responses in order to prevent certain types of crashes/incidents. This predominant response of respondents comes with the obligation of drone builders to consider creating the possibility of initiating responses of the device as a measure to prevent unpleasant events during negligent flight.

**Table 2.** Question responses—preventive measures for negligent flying damages.

| Response/Scale | 1 | 2 | 3 | 4 | 5 | Medium Score |
|---|---|---|---|---|---|---|
| Automatic actions taken by the drone to prevent incidents/accidents | 11 | 11 | 23 | 54 | 181 | 4.37 |
| Recurring training to be performed by the drone users | 14 | 17 | 44 | 47 | 157 | 4.13 |
| Getting a driving license after significant training as a drone driver | 23 | 22 | 32 | 50 | 152 | 4.03 |

Source: Author's processing.

Thus, most respondents consider that technology should play a prominent role in preventing negligence on the part of users. This entails additional costs and research on the part of drone producers. However, it entails that respondents also view drone producers as responsible for the prevention of accidents, even in such situations. This can also be reflected through the mechanisms mentioned above and in Section 5 of this paper.

Further, the second option considered by the respondents is the one referring to the continuous training of users. This has a higher score than the initial hands-on training. This means that the respondents' views are reflected accurately in the proposals made in this paper in terms of continuous hands-on training and awareness for users. This is the only manner to keep the users updated with recent threats and to keep them informed about prevention techniques such as setting an altitude for RTH-return to home command, and upgrade to the existing operating system version to the latest release should be made in order to operate the drone in safely and reliable mode. Further, from a technical perspective, multiple methodologies can be applied to prioritize in a risk-based approach given the wide area of external factors with which drones interact with a multitude of instructions received from drone users. In this case, the various sensor data analyzed should have in place specific checks to analyze the benchmark values.

This is correlated with the proposals made in the discussions section, which emphasize exactly the continuous nature of accident prevention.

Further, *this invalidates H2*, as the periodical training of drone users is considered useful with a score of 4.13; however, the most effective method is considered to be automatic responses embedded into the drone software to prevent such negligence of the user (with a score of 4.37). Therefore, the respondents view that periodic training may be useful, but is not sufficient to stop negligence and should be coupled with automatic responses embedded into the software (e.g.,: an RTH-return to home command is very useful in order to mitigate the loss of communication of the user with the drone)

These responses also contribute to clarifying H2. Respondents view the need for technical and organizational measures to be taken by all stakeholders involved in the production/distribution/use of drones (including drone users) in order to prevent negligence incidents and damages. This entails that the respondents consider the additional costs and time-consuming actions as necessary and are willing to bear the additional costs in view of ensuring safer drone usage.

*H5 is validated*, as the users prefer to have an automated mechanism rather than relying on manual actions to be taken by users. This emphasizes the need for reliable and resilient automation in view of obtaining the trust of customers for a specific product.

*One of the main principles in cyber-attack prevention is that of cyber awareness and training. To this end, both the draft NIST SP 800-82 and the draft Cyber Resilience Act mention the need for training and information for users. Although this can diminish certain types of cyber-attacks, as indicated by the responses to the questionnaire, additional automatic safety features may be developed for drones in order to ensure limitation of damages in case of negligence of the users when flying the drone.*

*Thus, the proposal for this use case is the inclusion of obligation for the stakeholders to analyze and develop safety features for the negligence of users. This type of clarification can be tied with the vulnerability management one mentioned in both NIST SP 800-82 and the Cyber Resilience Act throughout the lifetime of the drone.*

### 5.2. Malfunctioning of Software or Hardware

Malfunctioning entails the lack of a proper algorithm (e.g., insufficiently tested, having bugs not identified during the quality assurance phase) or proper hardware component creation/integration, without intervention from threat actors [33].

One question from the questionnaire was "Who is responsible when a hardware malfunction occurs?". The answer variants can be consulted in the legend of Figure 4.

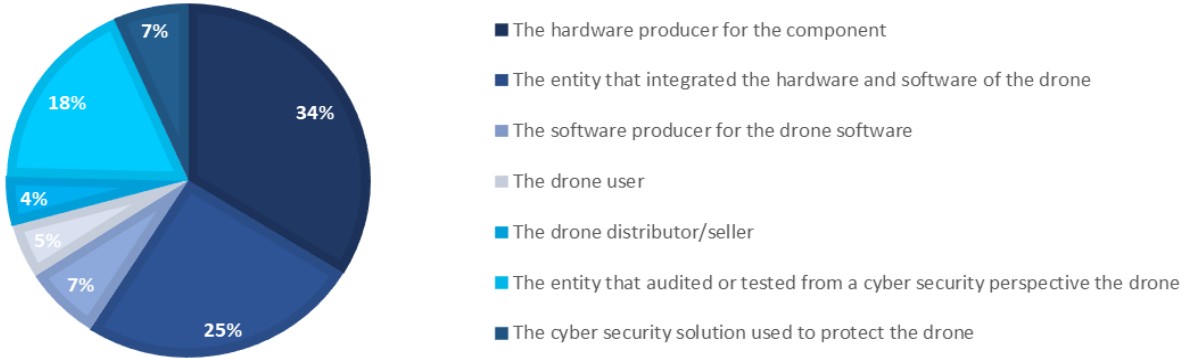

**Figure 4.** Question responses—malfunctioning of hardware components. Source: Author's processing.

As can be seen in Figure 4, 34% of respondents believe that the hardware producer is to blame for the malfunctioning drone hardware components. However, 25% of respondents believe that those who integrate software and hardware components into drones could be responsible. Situations in which distributors, sellers or users are responsible for such inconveniences are exceptions.

Therefore, for the hardware component, it seems that the opinions of the respondents are split between the hardware producer (34%) and the integrator (25%), with the majority of such respondents located in Europe. This is in line with the view of the respondents

on the software vulnerabilities as well. These should be identified by the producer (or, in this case, integrator). It is interesting that the distributor/seller has only 4%, which is significantly less than the case of the software vulnerabilities question.

*H3 is invalidated*. The integrator was mentioned with only 25% of the responses, whereas the hardware producer received the highest percentage (34%). As mentioned, this is debatable from a technical perspective, as certain vulnerabilities can be identified by the hardware producer and certain vulnerabilities by the integrator when testing the interaction of the hardware with its software.

*H4 is validated*, as the focus of the responses is on the hardware producer, the entity that has incorporated the drone components and the entity that audited the drone before entrance on the market. It is interesting to see that the software producer itself only received 7% of the responses, whereas the hardware producer received 34% and the integrator 25%.

These results are in line with the provisions of the draft NIST SP 800-82 (Section 5.2.5 Layer 5—Software security of NIST SP 800-82) and the draft EU Cyber Resilience Act (for example, in article 10), which focus on the producers, but also on the importers and distributors. *What may be worth noting for this use case is that the other stakeholders aside from the producers may need to have a more active role in the vulnerability management than currently provided in the draft EU Cyber Resilience Act and in the draft NIST SP 800-82.*

Another interesting question in terms of addressing vulnerabilities and prevention of vulnerabilities was "Who is responsible if vulnerabilities are found in the drone software from the beginning and these allow an attacker to take over the drone?" with responses as presented in Figure 5 below.

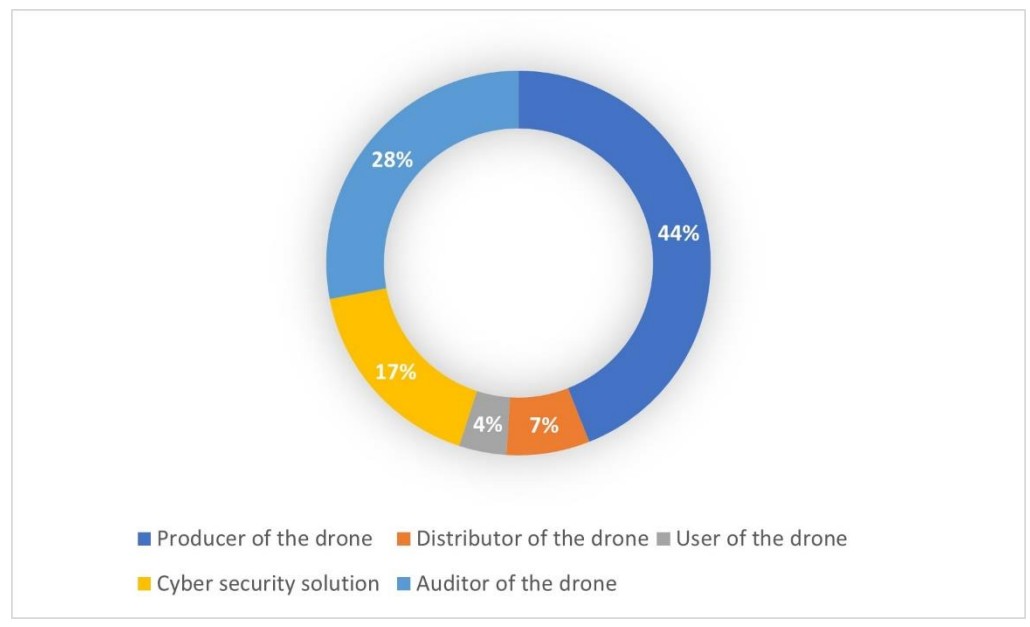

**Figure 5.** Question—liability in case of outset vulnerabilities. Source: Author's processing.

The question includes additional insight into the view of respondents in terms of stakeholder best placed to implement measures to prevent or address vulnerabilities. This details additionally the view of the respondents on the lifecycle of drones and the fact that some stakeholders should be involved in multiple steps of the drone lifecycle.

It is interesting that respondents consider the drone software producer the entity that can find out at the beginning (before distribution of the drones on the market), but also subsequently, until the drone is not proper for use, the vulnerabilities in the drone software at the outset (and the updates offered by the drone software producer during the lifecycle of the drone).

*H6 has been validated* in the sense that the drone software producer is viewed as the one best placed to identify and mitigate vulnerabilities, as this entity hold the most information

about the proprietary drone source code and can perform various periodical tests in this respect over the lifetime of the drone. It is interesting to see, however, that the cyber security solution provider and the auditor are viewed as gatekeepers of the vulnerability identification and mitigation, as independent parties guaranteeing the safety and security of the drone for use. Thus, the two stakeholders have a cumulated 45%, with the majority of such respondents located in Europe.

One question was "What option is the most relevant preventive measure when vulnerabilities are found in the initial software of the drone? (one to five scale, one meaning total disagreement and five meaning total agreement), with responses as detailed in Table 3.

**Table 3.** Question–preventive measures for initial vulnerabilities.

| Response/Scale | 1 | 2 | 3 | 4 | 5 | Medium Score |
|---|---|---|---|---|---|---|
| Auditing from a cyber security perspective before drones are distributed on the market | 7 | 9 | 31 | 41 | 165 | 4.38 |
| Recurrent cyber security auditing on the drone | 34 | 26 | 52 | 45 | 96 | 3.57 |
| Failsafe mechanisms in case attackers have taken over the drone in order to land the drone safely | 14 | 9 | 30 | 48 | 152 | 4.25 |
| Cyber security software on the drone for detection and prevention | 6 | 8 | 25 | 37 | 177 | 4.47 |

Source: Author's processing.

This question is closely tied to the previous one. The preventive steps to identify and address vulnerabilities also reflect the stakeholder best placed to perform such tasks.

Three options are very close to one another. This shows, on the one hand, the fact that respondents view the need of multiple layers of preventive measures to ensure proper handling of vulnerabilities. On the other hand, the respondents view that most stakeholders can contribute to this aspect in various moments of the drone lifecycle. For instance, the auditing performed at the moment when the drone is placed on the market is viewed as very important, at 4.38, whereas periodical auditing is viewed as less effective, at 3.57.

Further, the automation of cyber security is seen as a necessity by the respondents, with the highest score being granted to cyber security software and a high score also to automatic failsafe mechanisms.

This partially invalidated H7, as only the automated cyber security solution, had the highest score, not the periodical auditing.

*Thus, the obligations mentioned in standards such as the draft NIST SP 800-82 and draft EU Cyber Resilience Act should also have this in mind when detailing the aspects concerning vulnerability management, as only limited publishing or making available of vulnerabilities is not considered sufficient by the expert in the field. To this end, the above aspects can be reflected in Sections 5.2.5, 6.4.3 and 6.2.4.5 of the draft NIST SP 800-82 and in Article 10 of the draft Cyber Resilience Act.*

*Further, this indicates that the conformity assessment mentioned under the draft Cyber Resilience Act is not considered by experts in the field as sufficient to prevent damages and cyberattacks. Rather, enhanced automation on the side of the producers and the other entities that are stakeholders in the drone ecosystem is needed, thus providing a real-time response to prevent malfunctioning and exploitation of vulnerabilities.*

## 6. Discussions

This section addresses the context from a standardization, technical process framework and technical perspective for the above results and provides additional background on potential approaches to prevention mechanisms, including from a standardization and technological process organizational perspective.

### 6.1. User Actions Resulting in Economic Damages

For certain types of flights of drones, security procedures have to be set in place by the user of the drone. These may be considered applicable for flight trajectory or other similar aspects [34].

Economic damages towards individuals or third-party objects on the ground (negligence or intent of user in flying the drone), which can include damages to privacy of individuals on the ground, damages in case of crashing drones [35].

Damages towards other drones can occur, including drones crashing in the air or other objects in the air (negligence or intent of the user in flying the drone).

For these types of damages, given the intent with which the action is performed, generally, the user is liable for the actions and damages incurred [36]. There may be situations in which the users act with negligence and their actions may result in damages. In such cases, it may be debated whether the drone software included sufficient security measures to prevent the actions of the users that can lead to damages (e.g., acceleration when an obstacle was detected) or if the training/user manual was sufficient to explain to the user the consequences of certain commands or the steps to be taken to perform a specific command.

Although the liability in this case may be more clearly established under the existing legal requirements, in order to prevent such damages from occurring, the current legislation does not include express provisions [13].

Recommendations for this use case are on the approaches for prevention mechanisms. One approach is the training for certification, which can include practical flying and knowledge of legal consequences of certain actions or damage generation, which is a mechanism for awareness building. Another approach concerns technical safety mechanisms for certain flight maneuvers. This includes, for instance, automatically avoiding obstacles in order to prevent collisions.

These types of measures entail time and costs on the part of drone producers, distributors, users and other stakeholders involved in the certification process. Nevertheless, given the novelty of the field and the lack of standardization in drone specifications across producers or standardization requirements across countries, such preventive measures may aid in reducing damages generated by drone users, either by intent or negligence.

Thus, from a standardization standpoint, the maturity level of the standardization and of the technological process framework in the drone field can be improved by addressing aspects that are already being encountered in practice and by placing an emphasis on prevention mechanisms in order to ensure public and personal safety. This point can improve both the draft NIST SP 800-82 standards and the draft EU Cyber Resilience Act in the sections mentioned above.

Consequently, this use case refers to negligence of the user in giving permissions to attackers to take over the drone by negligence. This can be done in a series of manners, but mainly through malware installed on an IT system/device having a connection to the drone and spreading to the drone itself.

### 6.2. Malfunctioning of Software or Hardware

From a technical perspective, generally bugs can occur even when software has been thoroughly tested as per the best practices in place at a given point in time. In order to mitigate as much as possible, the risks of malfunctioning, continuous analysis should be performed (e.g., vulnerability assessment, bug hunting) in order to identify any such cases.

These aspects refer to the safety part of the drone, as mentioned above in the product safety legislation especially. Nevertheless, this can also have implications from a civil tort law perspective, provided all the conditions (inaction that leads to a malfunction and damage, existence of damages, liaison with inaction of an entity) are fulfilled [37].

As in the other use cases mentioned above, this generally entails a burden of proof on the users, lengthy litigation and lack of trust of users/the public in drone safety. Thus, from an economical perspective, in order to increase the use of drones in commercial actions,

certain preventive steps should be considered, together with clearer specific liability for stakeholders involved in the creation of the software/hardware of the drone and its maintenance. Further, clarifications in terms of entities (producers, distributors, maintenance entities) inside or outside the EU should be addressed as well, in order to ensure trust in the supply chain around drone distribution and use [38].

In terms of the current legislation, there needs to be a distinction between producers of drone components and of entities that place on the market the drones that are resolved through a liability of the latter, with possibility to recover damages from the former, if needed, either in the same litigation case or separately [39].

As certain malfunctioning can be identified years after drones are placed on the market (similar to the case of cars), obligations that can be contemplated by the legislator can include the continuous monitoring of the drones' conditions.

Further, in certain situations, drone users can be permitted to change the existing software or install their own software on the drone itself or on the ground control system. Such software modifications can create vulnerabilities, with causes varying from improper implementation of the software that can introduce backdoors or allow certain kinds of exploits to infected software that can give access to attackers to the UAV. Limitations in terms of possibility to make changes and prior verification of such changes before using the drone can be implemented as preventive security measures [40].

The potential manner of mitigating such risks can include a preventive measure, such as prior certification of drones before they are placed on the market and periodical verification thereof in terms of vulnerability identification based on specific tests in this respect [2]. The period can be established generally (e.g., once a year) and include emergency checks in case new types of bugs/malfunctioning are identified. Such testing for certification and recertification should be more extensive than the current technological checklist verification performed under the drone distribution legislation mentioned above. As mentioned in the previous sections, the current version includes references to best practices and state of the art, without setting clearly defined tests to be performed and key performance indicators. The certification aspect has been included in the draft Cyber Resilience Act in article 10. However, the scope of the certification detailed therein should be further extended to cover the aspects mentioned above on vulnerability management and automation mechanisms for prevention of cyber-attacks. As the drat NIST SP 800-82 does not detail such a certification mechanism, this can be included therein.

Since an important part of the drones (especially in the consumer segment) rely on a third party device to act as the ground control station, there are even a greater number of threats that can affect the functioning of the aircraft. Vulnerabilities introduced by the mobile phones or tables used to control such vehicles can be exploited so that attackers can get partial or full control of a drone. Thus, another way of verifying the normal functioning of the drone could be the use of antivirus software on the ground control station, which can detect common malicious routines and alert the drone user about their existence, possibly preventing the drone from being launched.

Moreover, the data link layer in the UAV architecture might be targeted for various attacks, including GPS jamming or DoS. A secured communication layer between the aircraft and the ground control system (e.g.,: using TLS) or usage of inertial navigation systems, which do not rely on external input, could be useful in preventing such types of attacks [41].

In the case of very serious and dangerous types of bugs/errors that can lead to great economic damages, one option can be for public announcements of the producers for customers to return the respective drone model having issues [42]. Furthermore, over-the-air communication mechanisms for locking the drone in place until the latest update packs are installed can become useful for preventing serious accidents.

As in the case of cars, this relies on the actions of the actual consumers/users and may not prove effective because the news may not reach the users. For this reason, direct and immediate communication processes can be set in place between the producers of drone

hardware/software and users (involving relevant authorities or not). In addition, insurance for such malfunctioning may be considered as a transfer of risk to some extent [43].

These are options to reflect the responses in the questionnaire and can be adjusted based on the supply chain mechanisms in place and on the industry framework, depending on the markets taken into account as well. From a technological process perspective, a risk-based approach can be designed by the relevant stakeholders, each for their part in the lifecycle of the drone. To this end, from a technical perspective, methods such as threat modeling or estimation of likelihood of certain types of attacks through a red teaming exercise analysis can be taken into account, as detailed in [44,45]. Examples of such types of risk-based analysis can be found in [46]. Nevertheless, this type of example has to be further developed based on the technical context known by the relevant stakeholder.

The risk management aspects are included in the draft NIST SP 800-82, as well as the supply chain analysis (Sections 4.2.1, 6.1.3, 6.1.4 and 6.1.5 of NIST SP 800-82). Nevertheless, these are viewed more from the purchaser perspective rather than the other stakeholders in the drone ecosystem. The added value of the above proposal entails the continuous/periodical risk assessment and risk management performed by the stakeholders in the drone ecosystem.

The draft EU Cyber Resilience Act provides under article 10(5) an obligation for producers to perform and update risk assessments for the products they produced. The risk assessment also includes risks resulting from third-party components included in the drone software by the producer. Nevertheless, the additional step of taking mitigation measures to address the identified risks and reduce the likelihood or impact of such risks should also be included [47]. Further, in certain cases, the producer is not the best placed in the drone ecosystem to perform the risk assessment [48]. Thus, additional obligations of other stakeholders involved in the adjusting or modifying the drone should be included.

The above two measures can lead to additional costs being incurred by producers of hardware or software of drones and, in turn, by users of drones. The actual implementation of these measures should be tailored in order to minimize costs, while, concurrently, ensuring a high level of security in a timely manner for drone users. In this case, the damages avoided through the preventive measures have to be taken into account, together with the time and costs of litigation. This results in a cost-benefit analysis of preventive measures, which is in line with other tendencies in European legislation, such as the one concerning medical devices (Regulation (EU) 2017/745) and essential services (NIS Directive and draft NIS Directive 2.0) [49,50].

### 6.3. Summary of Proposals

Clarity on liability in case of negligence is required, based on certain objective and subjective criteria, including types of action/inaction that had to be performed by the drone user, context of the flight and of the drone and training attended by the drone user. This clarity can be reflected in a series of manners, including through a standardization process at the industry level or through the cooperation mechanisms between the drone stakeholders. The aim of this liability establishment within this paper is to identify the best placed stakeholders for implementing or ensuring the implementation of prevention mechanisms in case of negligence and malfunctioning.

Prevention mechanisms that can be required by standardization, by technological process frameworks or had in mind by producers are the automatic failsafe mechanisms and periodical training. The failsafe mechanisms entail that, in certain cases (e.g., swift dropping of altitude, user commands that are not appropriate for the type of flight area or typical for the user), the drone can enter an automatic flight mode and land safely. The periodic training ensures that drone users are updated on the latest techniques for safe flying. In terms of draft NIST SP 800-82 and draft EU Cyber Resilience Act, these points can be included, as these have not been considered to date.

In order to prevent malfunctioning of drones, an assessment thereof before placing them on the market together with periodical assessments may be useful. In addition,

sensors and key performance indicators may be useful to notify the drone users of potential issues or to set-off automatic prevention mechanisms in the drone software. This reflects the organizational process in terms of the drone lifecycle that has technical implications concerning the assessment of the drone and the creation/amendment of the drone software and hardware. Such aspects can be established at the industry level among the players on that market in terms of standardization with the aim of ensuring safety in the products placed on the market. In terms of the draft NIST SP 800-82 and draft EU Cyber Resilience Act, these points can improve the draft provisions therein.

The role of each component producer has to be determined on a case by case basis for the initial and periodic assessment and to monitor the possibility of malfunctioning and to embed automatic prevention mechanisms if possible.

Thus, in terms of negligence, from a prevention perspective, automatic mechanisms are the ones considered by respondents as most efficient, rather than training pilots and relying on this as a prevention mechanism. This includes additional pressure on the conditions to be fulfilled by the stakeholders creating the software and hardware of the drone, including the maintenance services for the drone.

Further, the stakeholders early on in the drone lifecycle are considered best placed to ensure proper organizational and technical measures are in place. This can be reflected in standards and frameworks in this respect, including codes of conduct.

In addition, automatic preventive measures are considered preferred to manual ones or to periodic revisions/auditing. This increases the pressure of automation and swift updating on the stakeholders involved in the drone's lifecycle and may prove difficult to implement in practice unless proper standardization and frameworks are included in this respect. In terms of the draft NIST SP 800-82 and draft EU Cyber Resilience Act, these points can improve the draft provisions therein, especially in terms of proactive actions to be taken and the roles of various/multiple stakeholders in the drone ecosystem, not only the producer.

*6.4. Limitations and Future Directions of the Research*

As detailed above, the scope of this research paper is to analyze the view of the individuals working daily with drones in various contexts and relevant angles (IT, legal, etc.) in terms of the existing standardization related to prevention of damages and other negative consequences in the case of the negligence of users and malfunctioning.

In terms of legislative norms, the current legislation at the EU level concerning drones has focused on the flying limits (areas where drones can fly, height of flight) and flight lessons to be taken by the individual flying the drone. Matters such as liability and preventive organizational and technical measures are currently handled more at the national level based on prior general civil law legislation. Further, standards or frameworks at the industry level on these two topics are limited.

For these reasons, the view of relevant experts on how such topics should be addressed in terms of standardization/framework is essential to be taken into account in further standardization on the topic. This brings the element of novelty to the article as it represents a summary of deficiencies in current standardization and proposed remediation and prevention approaches.

Thus, this research paper does not aim to propose changes to legislation and does not target other markets than the EU. Further, the aim is not to establish liability for certain particular situations, but, rather, to identify the view of the respondents in terms of the best placed stakeholder in the drone ecosystem to address preventive measures for certain use cases, and the negligence of users and malfunctioning.

In future articles related to drones, we aim to further research the topics in this paper and to make a comparison, as applied and grounded as possible, regarding the differences concerning the legislation and the exploitation of drones in different regions of the globe—in comparison with the European legislation. This type of research can lead to a standardization of the rules and measures that can be implemented and can eventually be

constituted as a set of proposals that can be submitted for study, evaluation and approval, both to the specialized commissions within the International Telecommunication Union, as well as other international institutions qualified in the field.

## 7. Conclusions

This research paper has focused on two main areas that generate damages without willful misconduct and has detailed the manner in which the liability for such actions can be interpreted in the drone ecosystem, with emphasis also on the prevention mechanisms that may be available. The aim for the liability aspect is to analyze the manner in which customers view the roles of each stakeholder in the drone lifecycle. This viewpoint is useful for the next step of the research, which entails corroborating the liability perspective with the positioning of the entity to take preventive measures for malfunctioning and negligence of users. The preventive measures analyzed in this research paper are on a technical and organizational level, which can entail specific processes to be implemented at industry level or standardization level and, subsequently, specific technical aspects to be implemented.

As technology is ever changing, so is the manner in which liability is shaped and analyzed, taking into account all stakeholders throughout the product/service lifecycle. In the case of drones, we have seen interesting responses from the questionnaire respondents in terms of liability and stakeholder best placed to prevent certain unintentional damages.

In terms of negligence of the user resulting in negative consequences, 70% of respondents view the user as the responsible person (validating H1), whereas 30% of the respondents view the producer (or integrator) of the drone as the responsible entity. This reflects the view of respondents (having the highest score in terms of preventive measures—4.37 and invalidating H2 which mentioned periodic training) that certain security feature should be embedded in the drone from the outset (e.g., automatic responses to prevent crashes/incidents) that help to prevent situations in which users act with negligence (e.g., they do not respect the use instructions, they do not exercise their duty of care).

In terms of hardware security, H3 was invalidated, as the respondents viewed the hardware producer as primarily liable (34%) and not the integrator (25%). In addition, H4 was validated, as stakeholders early on the drone lifecycle were considered responsible for setting in place proper preventive measures against malfunctioning and/or user actions, starting from the producers of components and up to the entity auditing the drone before it is released on the market. This also shows the important role of independent stakeholders verifying certain security and safety features of products. From a technical and organizational perspective, this entails the need for standardized proper verification mechanisms to be implemented by independent third parties, which can entail a black-box analysis or a grey box analysis (or even a white-box analysis). A grey or white box analysis can lead, from a technical perspective to a more comprehensive analysis and results reflecting the actual full status of the drone at that point in time.

Further, it shows that customers primarily focus on the moment the drone is produced and less on the maintenance needs throughout the lifecycle of the drone. This may be an interesting topic to explore further in terms of the reasoning of customers. Their response may relate to the fact that it is essential to properly formalize the placement on the market, whereas the maintenance represents a day-to-day activity. Or, alternatively, it may relate to the fact that customers view new vulnerabilities or prevention mechanisms not occurring during the lifetime of the drone, even though this scenario is highly probable given technological changes and vulnerability identification trends. From a technical perspective, this process is essential throughout the lifecycle of the drone, not just at the outset, when the drone is placed on the market. Thus, from a technical and organizational perspective, it is highly debatable to concentrate more on the placing on the market moment and less on the lifecycle of the drone.

In terms of H5, respondents viewed the automation process essential for preventing damages, with the role of the user being secondary in this case and an off-the-shelf solution being the desired approach in terms of preventive measures, without specific configuration

or actions from users. This entails that the producers of drone software and hardware are expected to work together towards anticipating negligence actions from the part of users and embed in the drone mitigating technical steps in this respect. Further, it is expected that the industry, through cooperation and standardization, is able to create a technical and organizational manner for avoiding malfunctioning in a continuously evolving vulnerability environment.

The research article has analyzed aspects in terms of technological process, need for standardization of the technical and organizational concerns related to the scope of this article from a prevention perspective. *Further, specific changes to draft standards in the field have been indicated in order to enhance the manner in which prevention mechanisms are addressed in a proactive and continuous manner, while involving all relevant stakeholders in the actions, not just the producer.*

Future research on the scope of this article can go in-depth on the recommendations in this article and develop a detailed layer of recommendations from an organizational and technical process framework, providing a taxonomy thereof.

The research in this article addresses only civil drones and not military or institutional drones and takes into account for analysis the current civil legislation at the level of the European Union and in civil law principles in such countries.

**Author Contributions:** Conceptualization, M.C.Ș. and L.G.; Methodology, M.C.Ș., M.V.A. and L.G.; Formal analysis, L.G., V.L.V., A.L.V. and L.A.; Investigation, M.C.Ș., L.G. and A.L.V.; Resources, V.L.V. and L.A.; Data curation and analysis L.G., V.L.V. and L.A.; Writing original draft preparation, L.G., V.L.V. and L.A.; Writing review and editing, M.C.Ș., M.V.A., L.G., V.L.V.,A.L.V. and L.A.; Visualization, M.C.Ș., M.V.A., L.G., V.L.V., A.L.V. and L.A.; Supervision, M.C.Ș.; Project administration, M.C.Ș.; Funding acquisition, M.C.Ș., V.L.V. and M.V.A. All authors have read and agreed to the published version of the manuscript.

**Funding:** This work was supported by a grant from the Romanian Ministry of Education and Research, CNCS-UEFISCDI, project number PN-III-P4-ID-PCE-2020-2174, within PNCDI III. This paper was co-financed by The Bucharest University of Economic Studies during the PhD program.

**Informed Consent Statement:** Informed consent was obtained from all subjects involved in the study.

**Data Availability Statement:** The data used in this analysis are not public, but available upon request.

**Conflicts of Interest:** The authors declare no conflict of interest. The funders had no role in the design of the study; in the collection, analyses or interpretation of data; in the writing of the manuscript, or in the decision to publish the results.

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
