# Peer review of "Proposals of Processes and Organizational Preventive Measures against Malfunctioning of Drones and User Negligence"

_drones, doi:10.3390/drones7010064_

Round 1

Reviewer 1 Report

Please extend your future work in analysing and in terms of EU (international) legislation! The article offers a strong base to find a suitable legislation environment to prevent drone incidents.

Author Response

Thank you for your recommendation. We plan to further extend this research and will include the direction you have proposed and have adjusted the Limitations of research and future directions of research section accordingly. 

Reviewer 2 Report

I think the research field of this manuscript should belong to social science, while this journal belongs to natural science. Although the research object of this manuscript is the drone, its specific research content is not related to drone research, but related rules of drone use. Therefore, I think this manuscript is unsuitable for publication in this journal.

Author Response

Thank you for your comment. We have chosen this paper as it relates to drones specifically and as it includes a multi-disciplinary approach which includes also administrative perspective (legislation, standards, etc). For this reason, we have considered this journal as suitable for our paper.

Reviewer 3 Report

The analysis should be completed by adding:
- The authors indicated surveying 233 respondents indicating the geographical breakdown on the map (Figure 2). The distribution of respondents between countries/areas should be completed (made more specific). This is because the answers given to the questions may be related to the internal rules in a given country,
- Information about the sample size (233 respondents) in the context of statistical inference,

Author Response

Thank you for your recommendation. We have included additional insights on the respondents and the geographical breakdown for the limitations to be much clearer. In future studies we intend to have a comparison, as applied and grounded as possible, with respect to the geographical differences and how these can be addresses given their context.

Reviewer 4 Report

I'm reviewing this article for the umpteenth time and I note the large amount of work the authors have put into improving the article. As it stands, the article has the correct structure and, in my opinion, can be a valuable source of information for those preparing regulations in the area of drone use. There are minor errors in the article, e.g. 

- The same line concerning the FAA appears twice in Table 1.

- Figure 1 has a very poor resolution 

- I would ask the authors to indicate the keywords and databases they used for articles search  in literature overview  (e.g.: MDPI, Google scholar, IEEE...).

In conclusion, I believe that the article with minor corrections can be published . 

Author Response

Thank you for your recommendations. We have addressed these in the updated version (we have included reference to web of science database). Thank you especially for the recommendation concerning multiple databases. We will have this is mind for future researches.